# STNet: Spectral Transformation Network for Solving Operator Eigenvalue Problem

## Abstract

Operator eigenvalue problems play a critical role in various scientific fields and engineering applications, yet numerical methods are hindered by the curse of dimensionality. Recent deep learning methods provide an efficient approach to address this challenge by iterative updating neural networks. These methods' performance relies heavily on the spectral distribution of the given operator: larger gaps between the operator's eigenvalues will improve precision, thus tailored spectral transformations that leverage the spectral distribution can enhance their performance. Based on this observation, we propose the **S**pectral **T**ransformation **Net**work (**STNet**). During each iteration, STNet uses approximate eigenvalues and eigenfunctions to perform spectral transformations on the original operator, turning it into an equivalent but easier problem. Specifically, we employ deflation projection to exclude the subspace corresponding to already solved eigenfunctions, thereby reducing the search space and avoiding converging to existing eigenfunctions. Additionally, our filter transform magnifies eigenvalues in the desired region and suppresses those outside, further improving performance. Extensive experiments demonstrate that STNet consistently outperforms existing learning-based methods, achieving state-of-the-art performance in accuracy.

## 1. Introduction

The operator eigenvalue problem is a prominent focus in many scientific fields (Elhareef & Wu, 2023; Buchan et al., 2013; Cuzzocrea et al., 2020; Pfau et al., 2023) and engineering applications (Diao et al., 2023; Chen & Chan, 2000). However, traditional numerical methods are constrained by the curse of dimensionality, as the computational complexity increases quadratically or even cubically with the mesh size (Watkins, 2007).

A promising alternative is using neural networks to approxi-

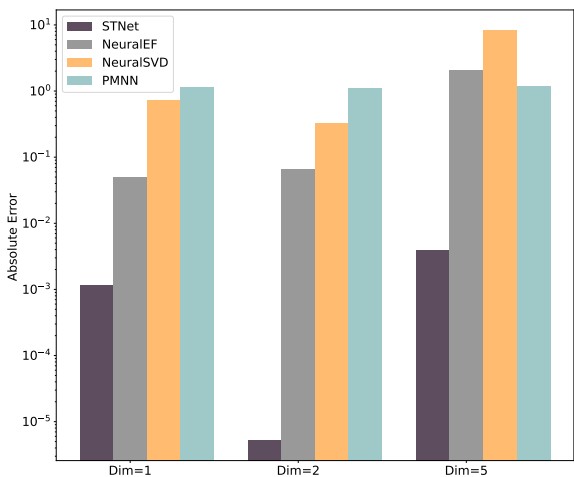

Figure 1: Absolute error results of eigenvalues for the Fokker-Planck operator computed using various algorithms, the x axis represents the operator dimension.

mate eigenfunctions (Pfau et al., 2018). These approaches reduce the number of parameters by replacing the matrix representation with a parametric nonlinear representation via neural networks. By designing appropriate loss functions, it updates parameters to approximate the desired operator eigenfunctions. These methods only require sampling specific regions without designing discretization mesh, significantly reducing the algorithm design cost and unnecessary approximation errors (He et al., 2022). Moreover, neural networks generally exhibit stronger expressiveness than linear matrix representations, requiring far fewer sampling points for the same problem compared to traditional methods (Nguyen et al., 2020).

Despite these advantages, the performance of such methods strongly depends on the operator's spectral distribution: if the target eigenvalues differs greatly to each other, the algorithm converges much more faster; otherwise, it may suffer from inefficient iterations. To improve convergence, spectral transformations can be designed based on the spectral distribution, reformulating the original problem into an equivalent but more tractable one. However, since the

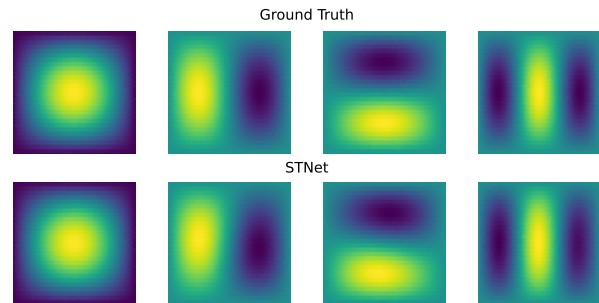

Figure 2: Comparison of the eigenfunctions of the 2D Harmonic operator computed by STNet and the Ground Truth.

real spectrum of the operator is initially unknown, existing approaches do not optimize spectral properties through such transformations.

To address this limitation, we propose the Spectral Transformation Network (STNet). By exploiting approximate eigenvalues and eigenvectors learned during the iterative process, STNet applies spectral transformations to the original operator, modifying its spectral distribution and thereby converting it into an equivalent problem that converges more easily. Concretely, we employ deflation projection to remove the subspace corresponding to already computed eigenfunctions. This not only narrows the search space but also prevents subsequent eigenfunctions from collapsing into the same subspace. Meanwhile, our filter transform amplifies eigenvalues within the target region and suppresses those outside it, promoting rapid convergence to the desired eigenvalues. Extensive experiments demonstrate that STNet significantly surpasses existing methods based on deep learning, achieving state-of-the-art performance in accuracy. Figure 2 presents the results obtained by STNet on the 2D Harmonic operator eigenvalue problem, alongside the ground truth, demonstrating our method's capability to accurately solve eigenvalue problems.

## 2. Related work

Recent advancements in applying neural networks to eigenvalue problems have shown promising results. Innovations such as spectral inference networks (SpIN) (Pfau et al., 2018), which model eigenvalue problems as kernel problem optimizations solved via neural networks. Neural eigenfunctions (NeuralEF) (Deng et al., 2022), which significantly reduces computational costs by optimizing the costly orthogonalization steps, are noteworthy. Neural singular value decomposition (NeuralSVD) employs truncated singular value decomposition for low-rank approximation to enhance the orthogonality required in learning functions (Ryu et al., 2024).

Another class of algorithms originates from optimizing the Rayleigh quotient. The deep Ritz method (DRM) utilizes the Rayleigh quotient for computing the smallest eigenvalues, demonstrating significant potential (Yu et al., 2018). Several studies have employed the Rayleigh quotient to construct variation-free functions, achieved through physics-informed neural networks (PINNs) (Ben-Shaul et al., 2023; 2020). Extensions of this approach include enhanced loss functions with regularization terms to improve the learning accuracy of the smallest eigenvalues (Jin et al., 2022). Additionally, Han et al. (2020) reformulate the eigenvalue problem as a fixed-point problem of the semigroup flow induced by the operator, solving it using the diffusion Monte Carlo method. The power method neural network (PMNN) integrates the power method with PINNs, using an iterative process to approximate the exact eigenvalues (Yang et al., 2023) closely. While PMNN has proven effective in solving for a single eigenvalue (Yang et al., 2023), it has yet to be developed for computing multiple distinct eigenvalues simultaneously.

Furthermore, in the field of computational chemistry, research on specialized model architectures for specific operators, such as the Hamiltonian, focuses on developing novel neural network ansatzes (Carleo & Troyer, 2017; Schütt et al., 2017; Choo et al., 2020; Pfau et al., 2020; Hermann et al., 2020; Gerard et al., 2022; Hermann et al., 2023). These architectures are designed to embed physical inductive biases better, enhancing expressivity. Additionally, there are studies employing neural networks for Quantum Monte Carlo (QMC) methods to tackle related problems in quantum chemistry (Cuzzocrea et al., 2020; Entwistle et al., 2023; Pfau et al., 2023).

## 3. Preliminaries

### 3.1. Operator Eigenvalue Problem

We primarily focus on the eigenvalue problems of differential operators, such as $\frac{\partial}{\partial x} + \frac{\partial}{\partial y}, \Delta$, etc. Mathematically, an operator $\mathcal{L} : \mathcal{H}_1 \to \mathcal{H}_2$ is a mapping between two Hilbert spaces. Considering a self-adjoint operator $\mathcal{L}$ defined on a domain $\Omega \subset \mathbb{R}^D$, the operator eigenvalue problem can be expressed in the following form (Evans, 2022):

$$\mathcal{L}v = \lambda v \quad \text{in } \Omega, \tag{1}$$

where $\Omega \subseteq \mathbb{R}^D$ serves as the domain; $v$ is the eigenfunction and $\lambda$ is the eigenvalue. Typically, it is often necessary to solve for multiple eigenvalues, $\lambda_i, i = 1, \ldots, L$.

### 3.2. Power Method

The power method is a classical algorithm designed to approximate the eigenvalue of an operator $\mathcal{L}$ in the vicinity of a given shift $\sigma$. By applying the shift $\sigma$ (often chosen as an

approximation to the target eigenvalue), the original eigenvalue problem is effectively transformed into an equivalent problem for the new operator $(\mathcal{L} - \sigma I)^{-1}$. In each iteration, the current approximate solution is multiplied by this new operator, thereby amplifying the component associated with the eigenvalue closest to $\sigma$. This iterative procedure converges to the desired eigenvalue. The pseudocode is shown below (Golub & Van Loan, 2013):

---

**Algorithm 1** Power Method for the Operator $\mathcal{L}$

---

1: **Input:** Operator $\mathcal{L}$, shift $\sigma$, initial guess $v^0$, maximum iterations $k_{\max}$, and convergence threshold $\epsilon$.
2: **Output:** Eigenvalue $\lambda$ near $\sigma$.
3: $v^0 = v^0 / \|v^0\|$ .
4: **for** $k = 1$ to $k_{\max}$ **do**
5:     $v^k = p^k / \|p^k\|$ and solve $(\mathcal{L} - \sigma I) p^k = v^{k-1}$.
6:     **if** $\|v^k - v^{k-1}\| < \epsilon$ **then**
7:         $\lambda = \frac{\langle v^k, \mathcal{L} v^k \rangle}{\langle v^k, v^k \rangle}$ and **break**.
8:     **end if**
9: **end for**

---

In each iteration, solving the linear system $(\mathcal{L} - \sigma I) p^k = v^{k-1}$ is equivalent to applying the operator $(\mathcal{L} - \sigma I)^{-1}$ to $v^{k-1}$. Afterward normalizing $v^k$ helps maintain numerical stability. Convergence is typically assessed by evaluating the error $\|v^k - v^{k-1}\|$, ensuring that the final solution meets the desired accuracy. The fundamental reason for the convergence of the power method lies in the repeated application of $(\mathcal{L} - \sigma I)^{-1}$, which progressively magnifies the component of $v^k$ in the direction of the eigenfunction with eigenvalue closest to $\sigma$. For a more detailed introduction to the power method, please refer to the Appendix A.1.

### 3.3. Deflation Projection

The deflation technique plays a critical role in solving eigenvalue problems, particularly when multiple distinct eigenvalues need to be computed. Deflation projection is an effective deflation strategy that utilizes known eigenvalues and corresponding eigenfunctions to modify the structure of the operator, thereby simplifying the computation of remaining eigenvalues (Saad, 2011).

The core idea of deflation projection is to construct an operator $\mathcal{P}$, often defined as $\mathcal{P}(u) = \langle u, v_1 \rangle v_1$ where $v_1$ is a known eigenfunction. This operator is then used to modify the original operator $\mathcal{L}$ into a new operator:

$$\mathcal{B} = \mathcal{L} - \lambda_1 \mathcal{P}. \tag{2}$$

In $\mathcal{B}$, the eigenvalue $\lambda_1$ associated with $v_1$ is effectively removed from the spectrum of $\mathcal{L}$. Additional details on deflation projection can be found in Appendix A.2.

### 3.4. Filter Transform

The filter transform is widely used in numerical linear algebra to enhance the accuracy of eigenvalue computations (Saad, 2011). By constructing a suitable filter function $F(\mathcal{L})$, the operator $\mathcal{L}$ undergoes a spectral transformation that amplifies the target eigenvalues and suppresses the irrelevant ones. The filter transform can effectively highlight the desired spectral region without altering the corresponding eigenfunctions (Watkins, 2007). Further details on the filter transform can be found in Appendix A.3.

## 4. Method

### 4.1. Problem Formulation

We consider the operator eigenvalue problem for a differential operator $\mathcal{L}$ defined on a domain $\Omega \subset \mathbb{R}^D$. Our goal is to approximate the $L$ eigenvalues $\lambda_i$ near a given shift $\sigma$ and their corresponding eigenfunctions $v_i$, satisfying

$$\mathcal{L} v_i = \lambda_i v_i, \quad i = 1, 2, \ldots, L. \tag{3}$$

To achieve this, we employ $L$ neural networks parameterized by $\theta_i$. Each neural network $NN_{\mathcal{L}}(\cdot; \theta_i)$ maps the domain $\Omega$ into $\mathbb{R}$, providing an approximation of the eigenfunction $v_i$:

$$NN_{\mathcal{L}}(\cdot; \theta_i) : \Omega \to \mathbb{R}, \quad i = 1, 2, \ldots, L. \tag{4}$$

In order to represent both the functions and the operators numerically, we discretize $\Omega$ by uniformly randomly sampling $N$ points:

$$S \equiv \{\boldsymbol{x}_j = (x_j^1, \ldots, x_j^D) \mid \boldsymbol{x}_j \in \Omega, \, j = 1, 2, \ldots, N\}, \tag{5}$$

Correspondingly, each neural network $NN_{\mathcal{L}}(\cdot; \theta_i)$ output a vector $\boldsymbol{Y}_i \in \mathbb{R}^N$, which approximate the values of the eigenfunction $\tilde{v}_i(\cdot) = NN_{\mathcal{L}}(\cdot; \theta_i)$ at these sampled points:

$$\tilde{v}_i(\boldsymbol{x}_j) \equiv \boldsymbol{Y}_i(j), \quad i = 1, 2, \ldots, L, \quad j = 1, 2, \ldots, N. \tag{6}$$

The approximate eigenvalues $\tilde{\lambda}_i$ are then obtained by applying $\mathcal{L}$ to the computed eigenfunctions $\tilde{v}_i$:

$$\tilde{\lambda}_i \equiv \frac{\langle \tilde{v}_i, \mathcal{L} \tilde{v}_i \rangle}{\langle \tilde{v}_i, \tilde{v}_i \rangle}, \quad i = 1, 2, \ldots, L. \tag{7}$$

Here, the differential operator $\mathcal{L}$ acts on the functions via automatic differentiation. We iteratively update the neural network parameters $\theta_i$ using gradient descent, aiming to

minimize the overall residual. Specifically, we formulate the following optimization problem:

$$\min_{\theta_i \in \Theta} \frac{1}{N} \sum_{i=1}^{L} \sum_{j=1}^{N} [\tilde{v}_i(\boldsymbol{x}_j) - v_i(\boldsymbol{x}_j)]^2, \qquad (8)$$

, where $\Theta$ denotes the parameter space of the neural networks. This approach does not require any training data, as it relies solely on satisfying the differential operator eigenvalue equations over the domain $\Omega$. Finally, this procedure provides approximations $\tilde{\lambda}_i$ of the true eigenvalues $\lambda_i$, $i = 1, \ldots, L$.

### 4.2. Spectral Transformation Network

Inspired by the power method and the power method neural network (Yang et al., 2023), we propose STNet to solve eigenvalue problems, as shown in Figure 3. In STNet, we replace the function $v^k$ from the $k$-th iteration of the power method with $\tilde{v}_i^k(x) \equiv NN_{\mathcal{L}}(x; \theta_i^k)$, where each neural network is implemented via a multilayer perceptron. Since neural networks cannot directly implement the inverse operator $(\mathcal{L} - \sigma I)^{-1}$, we enforce $(\mathcal{L} - \sigma I)\tilde{v}^k \approx \tilde{v}^{k-1}$ through a suitable loss function. The updated parameters $\theta_i^k \rightarrow \theta_i^{k+1}$ then yield $\tilde{v}^{k+1} = NN_{\mathcal{L}}(x; \theta_i^{k+1})$. Algorithm 2 shows the detailed procedure of STNet.

Classical power method convergence is closely related to the spectral distribution of the operator, which is unknown initially and thus difficult to optimize against directly. However, as the iterative process starts, we can get additional information—such as already computed eigenvalues and eigenfunctions. Using these results for the spectral transformation of the original operator can greatly improve subsequent power-method iterations. In our pseudocode 2, we introduce two modules to enhance performance:

- **Deflation Projection** uses already computed eigenvalues and eigenfunctions to construct a projection that excludes the previously resolved subspace, preventing convergence to known eigenfunctions and reducing the search space.

- **Filter Transform** employs approximate eigenvalues to construct a spectral transformation (filter function) that enlarges the target eigenvalue region and suppresses others, boosting the efficiency of STNet.

#### 4.2.1. DEFLATION PROJECTION

Suppose we have already approximated the eigenvalues $\tilde{\lambda}_1, \tilde{\lambda}_2, \ldots, \tilde{\lambda}_{i-1}$ and their corresponding eigenfunctions $\tilde{v}_1, \tilde{v}_2, \ldots, \tilde{v}_{i-1}$. To compute the $i$-th eigenfunction, we focus on the residual subspace orthogonal to the subspace spanned by these previously computed eigenfunctions.

---

**Algorithm 2** Spectral Transformation Network

1: **Input:** Operator $\mathcal{L}$ over domain $\Omega \subset \mathbb{R}^D$, shift $\sigma$, number of sampling points $N$, number of eigenvalues $L$, learning rate $\eta$, convergence threshold $\epsilon$, maximum iterations $k_{\max}$.
2: **Output:** Eigenvalues $\tilde{\lambda}_i, \quad i = 1, \ldots, L$.
3: Uniformly randomly sample $N$ points $\{\boldsymbol{x}_j\}$ in $\Omega$ to form dataset $S$.
4: Randomly initialize the network parameters $\theta_i^0$, as well as the normalized $\tilde{v}_i$, and set $\tilde{\lambda}_i = \sigma, \quad i = 1, \ldots, L$.
5: **for** $k = 1$ **to** $k_{\max}$ **do**
6: $\quad \tilde{v}_i^k(\boldsymbol{x}_j) = NN_{\mathcal{L}}(\boldsymbol{x}_j; \theta_i^k), \boldsymbol{x}_j \in S$.
7: $\quad \mathcal{L}_i' = D_i(\mathcal{L}), \quad i = 1, \ldots, L$   // **Deflation Projection**
8: $\quad \mathcal{L}_i'' = F_i(\mathcal{L}'), \quad i = 1, \ldots, L$   // **Filter Transform**
9: $\quad \tilde{u}_i^k(\boldsymbol{x}_j) = \frac{\mathcal{L}_i'' \tilde{v}_i^k(\boldsymbol{x}_j)}{\|\mathcal{L}_i'' \tilde{v}_i^k(\boldsymbol{x}_j)\|}, \quad i = 1, \ldots, L$.
10: $\quad \text{Loss}_i^k = \frac{1}{N} \sum_{j=1}^{N} [\tilde{v}_i^{k-1}(\boldsymbol{x}_j) - \tilde{u}_i^k(\boldsymbol{x}_j)]^2, \quad i = 1, \ldots, L$.
11: $\quad \theta_i^{k+1} = \theta_i^k - \eta \nabla_{\theta_i} \text{Loss}_i^k, \quad i = 1, \ldots, L$   // **Parameter Update**
12: $\quad$ **for** $i = 1$ **to** $L$ **do**
13: $\quad\quad$ **if** $\text{Loss}_i^k < \epsilon_i$ **then**
14: $\quad\quad\quad \epsilon_i = \text{Loss}_i^k, \tilde{\lambda}_i = \frac{\langle \tilde{v}_i^k, \mathcal{L}\tilde{v}_i^k \rangle}{\langle \tilde{v}_i^k, \tilde{v}_i^k \rangle}, \tilde{v}_i = \tilde{v}_i^k$.
15: $\quad\quad$ **end if**
16: $\quad$ **end for**
17: $\quad$ **if** $\epsilon_i < \epsilon$ for all $i$ **then**
18: $\quad\quad$ Convergence achieved; **break**.
19: $\quad$ **else**
20: $\quad\quad$ Update deflation projection and filter function: $D_i, F_i, \quad i = 1, \ldots, L$.
21: $\quad$ **end if**
22: **end for**

---

The deflated projection is then defined as

$$D_i(\mathcal{L}) \equiv \mathcal{L} - \mathcal{Q}_{i-1} \Sigma_{i-1} \mathcal{Q}_{i-1}^*. \qquad (9)$$

Here $\mathcal{Q}_{i-1}$ maps each vector $(\alpha_1, \ldots, \alpha_{i-1}) \in \mathbb{R}^{i-1}$ to the function $\sum_{k=1}^{i-1} \alpha_k \tilde{v}_k$, thus reconstructing functions from the span of $\{\tilde{v}_1, \ldots, \tilde{v}_{i-1}\}$. $\mathcal{Q}_{i-1}^*$ is the adjoint of $\mathcal{Q}_{i-1}$. And $\Sigma_{i-1}$ is a diagonal operator that scales each $\tilde{v}_k$ by its corresponding eigenvalue $\tilde{\lambda}_k$.

By employing the deflation projection, the gradient descent search space of the neural network is constrained to be orthogonal to the subspace spanned by $\{\tilde{v}_1, \tilde{v}_2, \ldots, \tilde{v}_{i-1}\}$. This projection prevents the neural network output $NN_{\mathcal{L}}(\theta_i)$ from converging to the invariant subspace formed by known eigenfunctions, thereby enhancing the orthogonality among the outputs of different neural networks $NN_{\mathcal{L}}(\theta_1), \ldots, NN_{\mathcal{L}}(\theta_{i-1})$. On one hand, this reduction

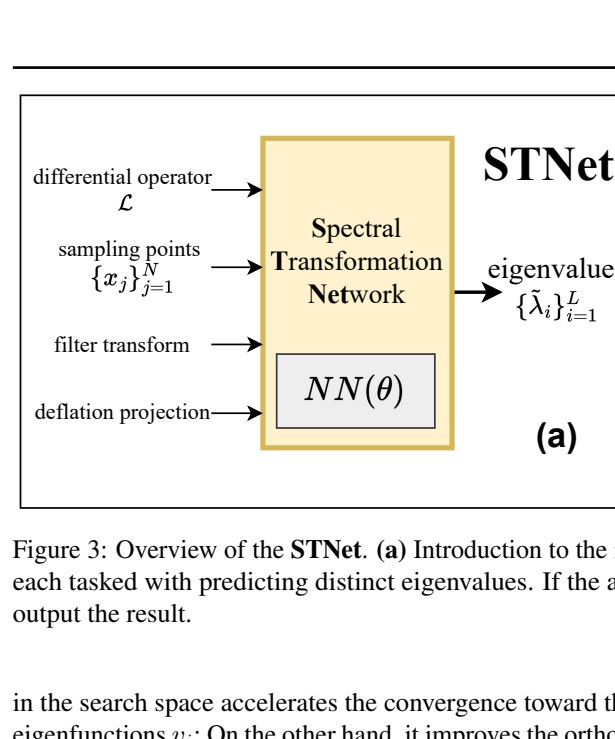

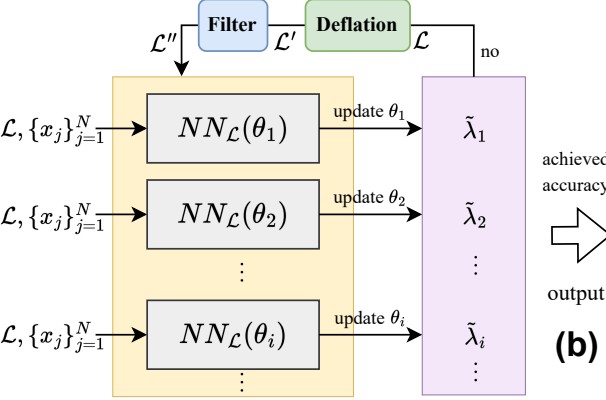

Figure 3: Overview of the **STNet**. **(a)** Introduction to the inputs and outputs. **(b)** STNet comprises multiple neural networks, each tasked with predicting distinct eigenvalues. If the accuracy of the solution reaches the expectation, then STNet will output the result.

in the search space accelerates the convergence toward the eigenfunctions $v_i$; On the other hand, it improves the orthogonality among the neural network outputs, which reduces the error in predicting the eigenfunction $\tilde{v}_i$.

In practice, we use the approximate eigenvalues and eigenfunctions with the smallest error in iterations to construct the deflation projection. This allows us to update adaptively, ensuring that the method remains effective when calculating more eigenfunctions.

### 4.2.2. FILTER TRANSFORM

During the iterative process, we can obtain approximate eigenvalues $\tilde{\lambda}_i$, and assume the corresponding true eigenvalues lie within $[\tilde{\lambda}_i - \xi, \tilde{\lambda}_i + \xi]$, where $\xi$ is a tunable parameter, typically $\xi = 0.1$ or $1$. We employ a rational function-based filter transform on the original operator to simultaneously amplify the eigenvalues in these intervals and thus improve convergence performance. Specifically, we transform

$$\mathcal{L} \longrightarrow \prod_{i_0=0}^{i-1} \left[ (\mathcal{L} - (\tilde{\lambda}_{i_0} - \xi)I)\,(\mathcal{L} - (\tilde{\lambda}_{i_0} + \xi)I) \right]^{-1}. \quad (10)$$

By contrast, the basic power method shift-invert strategy, $\mathcal{L} \to (\mathcal{L} - \sigma I)^{-1}$, can be viewed as a special case of this more general construction. In STNet, we simulate the inverse operator via a suitably designed loss function. Therefore, the corresponding pseudocode filter function $F$ removes the inverse, namely:

$$F_i(\mathcal{L}) = \prod_{i_0=0}^{i-1} \left[ (\mathcal{L} - (\tilde{\lambda}_{i_0} - \xi)I)\,(\mathcal{L} - (\tilde{\lambda}_{i_0} + \xi)I) \right]. \quad (11)$$

When $\lambda_i$ lies within $[\tilde{\lambda}_i - \xi, \tilde{\lambda}_i + \xi]$, the poles $\tilde{\lambda}_i \pm \xi$ make $\|F_i(v_i)\|$ sufficiently large for the corresponding eigenvector $v_i$. This repeated amplification causes that direction to dominate in the subsequent iterations, while eigenvalues outside those intervals are gradually suppressed. Consequently, the method converges more efficiently to the desired eigenvalues.

## 5. Experiments

We conducted comprehensive experiments to evaluate STNet, focusing on:

- Solving multiple eigenvalues in the Harmonic eigenvalue problem.

- Solving the principal eigenvalue in the Schrödinger oscillator equation.

- Solving zero eigenvalues in the Fokker-Planck equation.

- Comparative experiment with traditional algorithms.

- The ablation experiments.

**Baselines**: For these experiments, we selected three learning-based methods for computing operator eigenvalues as our baselines: 1. PMNN (Yang et al., 2023); 2. NeuralEF (Deng et al., 2022); 3. NeuralSVD (Ryu et al., 2024). In the comparative experiments with traditional algorithms, we chose the finite difference method (FDM) (LeVeque, 2007).

**Experiment Settings**: To ensure consistency, all experiments were conducted under the same computational conditions. For further details on the experimental environment

and algorithm parameters, please refer to Appendices B.1 and B.2.

## 5.1. Harmonic Eigenvalue Problem

Harmonic eigenvalue problems are common in fields such as structural dynamics and acoustics, and can be mathematically expressed as follows (Yang et al., 2023; Morgan & Zeng, 1998):

$$\begin{cases} -\Delta v = \lambda v, & \text{in } \Omega, \\ v = 0, & \text{on } \partial\Omega. \end{cases} \quad (12)$$

Here $\Delta$ denotes the Laplacian operator. We consider the domain $\Omega = [0,1]^D$ where $D$ represents the dimension of the operator, and the boundary conditions are Dirichlet. In this setting, the eigenvalue problem has analytical solutions, with eigenvalues and corresponding eigenfunctions given by:

$$\lambda_{n_1,\ldots,n_D} = \pi^2 \sum_{k=1}^{D} n_k^2, \quad n_k \in \mathbb{N}^+$$

$$u_{n_1,\ldots,n_D}(x_1,\ldots,x_k) = \prod_{k=1}^{D} \sin(n_k \pi x_k). \quad (13)$$

These experiments aim to calculate the smallest four eigenvalues of the Harmonic operator in $1, 2$ and $5$ dimensions. Since the PMNN model only computes the principal eigenvalue and cannot compute multiple eigenvalues simultaneously, it is not considered for comparison. NeuralEF, due to cumulative errors in its iterative orthogonalization process, experiences numerical instability in $2$ and $5$ dimensions, thus no data is available for these dimensions.

Firstly, as demonstrated in Table 1, the accuracy of STNet on all tasks is significantly better than that of existing methods. This enhancement primarily stems from the deflation projection. It effectively excludes solved invariant subspaces during the multi-eigenvalue solution process, thereby preserving the accuracy of multiple eigenvalues. This strongly validates the efficacy of our algorithm.

Secondly, in $5$-dimension, STNet consistently maintains a precision improvement of at least three orders of magnitude. As shown in Table 2, this is largely due to the STNet computed eigenpairs having smaller residuals (defined as $\|\mathcal{L}v - \lambda v\|_2$, see Appendix B.3 for details), indicating that STNet can effectively solve for accurate eigenvalues and eigenfunctions simultaneously.

Table 2: Residual comparison for eigenpairs of STNet and NeuralSVD for solving 5-dimensional Harmonic eigenvalue problems. The first row indicates the eigenpair index.

| Index | $(v_1, \lambda_1)$ | $(v_2, \lambda_2)$ | $(v_3, \lambda_3)$ | $(v_4, \lambda_4)$ |
|---|---|---|---|---|
| NeuralSVD | 5.924e+0 | 5.920e+0 | 5.921e+0 | 5.920e+0 |
| STNet | 4.864e-4 | 3.060e-3 | 5.980e-3 | 4.447e-3 |

Additionally, Table 1 reveals that in the process of solving multiple eigenvalues, the errors for subsequent eigenvalues tend to be significantly higher than those for earlier ones. NeuralEF and NeuralSVD exhibit relatively stable error change, and But STNet shows fluctuations (for instance, errors for $\lambda_2$ and $\lambda_3$ at dimension five are smaller than those for $\lambda_1$). This variability primarily arises because NeuralEF and NeuralSVD employ a uniform grid to acquire data points, whereas STNet uses uniform random sampling. While uniform random sampling inherently introduces some degree of randomness, it offers a significant advantage in high-dimensional settings. Specifically, a uniform grid necessitates an exponentially growing number of sampling points, $num^D$, where $num$ represents the number of grid points per dimension and $D$ denotes the operator dimension. In contrast, uniform random sampling is not subject to this constraint, making it more scalable for high-dimensional problems.

## 5.2. Schrödinger Oscillator Equation

The Schrödinger oscillator equation is a common problem in quantum mechanics, and its time-independent form is expressed as follows:

$$-\frac{1}{2}\Delta\psi + V\psi = E\psi, \quad \text{in } \Omega = [0,1]^D, \quad (14)$$

where $\psi$ is the wave function, $\Delta$ represents the Laplacian operator indicating the kinetic energy term, $V$ is the potential energy within $\Omega$, and $E$ denotes the energy eigenvalue (Ryu et al., 2024; Griffiths & Schroeter, 2018). This equation is formulated in natural units, simplifying the constants involved. Typically, the potential $V(x_1,\ldots,x_D) = \frac{1}{2}\sum_{k=1}^{D} x_k^2$ characterizes a multidimensional quadratic potential. The principal eigenvalue $E_0$ and corresponding eigenfunction $\psi_0$ are given by:

$$E_0 = \frac{D}{2}, \quad \psi_0(x_1,\ldots,x_D) = \prod_{k=1}^{D} \left(\frac{1}{\pi}\right)^{\frac{1}{4}} e^{-\frac{x_k^2}{2}}. \quad (15)$$

This experiment focuses on calculating the ground states of

Table 1: Absolute error comparison for eigenvalues of Harmonic operators. The first row lists the methods, the second row lists eigenvalue indices, and the first column lists the operator dimensions. The most accurate method is in bold.

| Method | NeuralEF | | | | NeuralSVD | | | | STNet | | | |
|---|---|---|---|---|---|---|---|---|---|---|---|---|
| | $\lambda_1$ | $\lambda_2$ | $\lambda_3$ | $\lambda_4$ | $\lambda_1$ | $\lambda_2$ | $\lambda_3$ | $\lambda_4$ | $\lambda_1$ | $\lambda_2$ | $\lambda_3$ | $\lambda_4$ |
| Dim = 1 | 1.4e-1 | 2.9e+1 | 7.9e+1 | 1.4e+2 | 1.0e-1 | 4.1e+1 | 1.0e+0 | 1.4e+2 | **6.3e-10** | **1.7e-1** | **6.3e-1** | **1.6e+1** |
| Dim = 2 | - | - | - | - | 5.5e-2 | 2.1e-1 | 1.5e-1 | 2.6e+1 | **1.0e-5** | **3.0e-2** | **6.8e-2** | **1.0e-1** |
| Dim = 5 | - | - | - | - | 2.5e-1 | 2.9e+1 | 2.9e+1 | 2.9e+1 | **2.3e-4** | **9.5e-5** | **6.2e-5** | **1.3e-3** |

Table 3: Absolute error comparison for the principal eigenvalues of oscillator operators. The first row lists the methods, and the first column lists the operator dimensions. The most accurate method is in bold.

| Method | PMNN | NeuralEF | NeuralSVD | STNet |
|---|---|---|---|---|
| Dim = 1 | 1.17e-6 | 2.57e-2 | 2.53e-2 | **3.62e-7** |
| Dim = 2 | 9.07e-5 | 7.55e-2 | 4.01e-1 | **2.35e-6** |
| Dim = 5 | 3.92e-1 | 3.97e-1 | 4.37e+0 | **3.23e-1** |

the Schrödinger equation in one, two, and five dimensions, i.e. the smallest principal eigenvalues.

Firstly, as shown in Table 3, the STNet achieves significantly higher precision than existing algorithms in computing the principal eigenvalues of the oscillator operator.

Furthermore, the accuracy of STNet surpasses that of PMNN. Both are designed based on the concept of the power method. When solving for the principal eigenvalue, the deflation projection loss may be considered inactive. This outcome suggests that the filter transform significantly enhances the accuracy.

### 5.3. Fokker-Planck Equation

The Fokker-Planck equation is central to statistical mechanics and is extensively applied across diverse fields such as thermodynamics, particle physics, and financial mathematics (Yang et al., 2023; Jordan et al., 1998; Frank, 2005). It can be mathematically formulated as follows:

$$-\Delta v - V \cdot \nabla v - \Delta V v = \lambda v, \quad \text{in } \Omega = [0, 2\pi]^D,$$

$$V(x) = \sin\left(\sum_{i=1}^{D} c_i \cos(x_i)\right). \tag{16}$$

Here $V(x)$ is a potential function with each coefficient $c_i$ varying within $[0.1, 1]$, $\lambda$ the eigenvalue, and $v$ the eigen-function. When the boundary conditions are periodic, there are multiple zero eigenvalues.

The eigenvalue at zero significantly impacts the numerical stability of the algorithm during iterative processes. This experiment investigates the computation of two zero eigenvalues for the Fokker-Planck equations with different parameters in 1, 2, and 5 dimensions. Due to the inherent limitation of the PMNN method, which can only compute a single eigenvalue, we restrict our analysis to calculating one eigenvalue when employing this approach.

As indicated in Table 4, the STNet algorithm significantly outperforms existing methods in computing the zero eigenvalues of the Fokker-Planck operator, effectively solving cases where the eigenvalue is zero. It is mainly due to the filter function, which performs a spectral transformation on the operator, converting the zero eigenvalue into other eigenvalues that are easier to calculate without changing the eigenvector.

### 5.4. Comparative Experiment with Traditional Algorithms

This experiment compares the accuracy of STNet and the traditional finite difference method (FDM) with a central difference scheme under identical point distributions ($6 \times 10^4$ points) (LeVeque, 2007). Both methods compute the four smallest eigenvalues of the 5D harmonic operator.

As shown in Table 5, STNet significantly outperforms FDM in accuracy. While FDM's precision depends on grid density, requiring exponentially more grid points and parameters with increasing dimensionality, STNet employs uniform random sampling instead of fixed grids. Leveraging neural networks' expressive power, STNet achieves higher accuracy with fewer parameters by effectively approximating eigenfunctions.

Traditional algorithms and neural network-based algorithms each have their own applicable domains. In low-dimensional scenarios, traditional algorithms significantly outperform neural network-based algorithms in terms of computational speed, and their accuracy can be improved by increasing

Table 4: Absolute error comparison for the principal eigenvalues of Fokker-Planck operators across algorithms. The first row lists the methods, the second row lists eigenvalue index, the first column lists the Fokker-Planck parameter and the second column lists the operator dimensions. The most accurate method is in bold.

| Method | | PMNN | NeuralEF | | NeuralSVD | | STNet | |
|---|---|---|---|---|---|---|---|---|
| $c_i$ | Dim | $\lambda_1$ | $\lambda_1$ | $\lambda_2$ | $\lambda_1$ | $\lambda_2$ | $\lambda_1$ | $\lambda_2$ |
| | 1 | 1.16e+0 | 4.98e-2 | 1.05e+0 | 7.19e-1 | 1.02e+0 | **1.17e-3** | **8.75e-3** |
| 0.5 | 2 | 1.11e+0 | 6.71e-2 | 1.57e+0 | 3.33e-1 | 1.03e+0 | **5.26e-6** | **5.14e-2** |
| | 5 | 1.17e+0 | 2.11e+0 | 9.17e+0 | 2.11e+0 | 4.82e+0 | **3.90e-3** | **1.29e-1** |
| | 1 | 8.60e-1 | 5.21e-1 | 5.95e-1 | 2.73e-1 | 3.19e-1 | **3.86e-2** | **2.33e-1** |
| 1.0 | 2 | 8.30e-1 | 6.58e-1 | 8.45e-1 | 2.75e-1 | 3.94e-1 | **1.99e-2** | **3.91e-2** |
| | 5 | 7.58e-1 | 7.71e-1 | 1.02e+0 | 2.01e-1 | 3.08e-1 | **5.64e-2** | **2.67e-2** |

Table 5: Absolute error comparison for eigenvalues of 5D Harmonic operators. The first column lists the methods, and the second column lists eigenvalue indexes

| Method | $\lambda_1$ | $\lambda_2$ | $\lambda_3$ | $\lambda_4$ |
|---|---|---|---|---|
| FDM | 4.05e-1 | 1.61e+0 | 1.61e+0 | 1.61e+0 |
| STNet | 2.31e-4 | 9.54e-5 | 6.21e-5 | 1.39e-3 |

the number of grid points. However, in high-dimensional problems, the number of required grid points grows exponentially with the dimensionality. For instance, while a 2D problem requires a $100^2$ grid, its 5D counterpart would need $100^5$ grid points. In such cases, enhancing accuracy by increasing the number of grid points becomes impractical. Neural network-based algorithms, on the other hand, offer an effective solution to these high-dimensional challenges.

**5.5. Ablation Experiments**

We conducted ablation experiments to validate the effectiveness of the deflation projection and filter transform modules. As shown in Table 6, the results for "w/o F" indicate that removing the filter transform significantly reduces solution accuracy. In the cases of "w/o D" and "w/o F and D," while the residuals remain small, the absolute errors for $\lambda_2$ and $\lambda_3$ are notably larger compared to $\lambda_1$. This suggests that without the deflation projection module, the network converges exclusively to the first eigenfunction $v_1$ corresponding to $\lambda_1$, failing to capture subsequent eigenfunctions. These findings underscore the critical roles of both modules: the filter transform enhances accuracy through spectral transformation. The deflation projection removes the subspace of already solved eigenfunctions from the search space, enabling the computation of multiple eigenvalues.

Additionally, experiments detailing the performance of STNet as a function of model depth, model width, and

Table 6: A comparison of different settings of STNet for the 2-dimensional Harmonic eigenvalue problem. "w/o" denotes the absence of a specific module, "F" represents the filter transform module, and "D" indicates the deflation projection module.

| | Index | $\lambda$ Absolute Error | Residual |
|---|---|---|---|
| STNet | $(v_1, \lambda_1)$ | 1.02e-5 | 4.12e-3 |
| | $(v_2, \lambda_2)$ | 3.04e-2 | 1.24e+1 |
| | $(v_3, \lambda_3)$ | 6.76e-1 | 1.43e+1 |
| w/o F | $(v_1, \lambda_1)$ | 6.73e-5 | 1.35e-2 |
| | $(v_2, \lambda_2)$ | 5.10e-2 | 4.72e+1 |
| | $(v_3, \lambda_3)$ | 1.06e-1 | 1.70e+2 |
| w/o D | $(v_1, \lambda_1)$ | 1.42e-5 | 4.12e-3 |
| | $(v_2, \lambda_2)$ | 2.96e+1 | 7.09e-3 |
| | $(v_3, \lambda_3)$ | 2.97e+1 | 1.09e-2 |
| w/o F and D | $(v_1, \lambda_1)$ | 6.73e-5 | 1.35e-2 |
| | $(v_2, \lambda_2)$ | 2.96e+1 | 1.45e-2 |
| | $(v_3, \lambda_3)$ | 2.97e+1 | 1.37e-2 |

the number of sampling points are provided in Appendix C.

# 6. Conclusions

In this paper, we present STNet, a novel learning-based approach for solving operator eigenvalue problems. By leveraging approximate eigenvalues and eigenvectors obtained during iteration, STNet employs spectral transformations to reformulate the original operator, altering its spectral distribution to create an equivalent problem with improved convergence properties. Experimental results show that STNet outperforms existing algorithms in accuracy across a wide range of operator eigenvalue problems.

## Impact Statement

This paper presents work whose goal is to advance the field of Machine Learning. There are many potential societal consequences of our work, none of which we feel must be specifically highlighted here.

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

## A. Background Knowledge and Relevant Analysis

### A.1. Convergence Analysis of the Power Method

Suppose $\boldsymbol{A} \in \mathbb{R}^{n \times n}$ and $\boldsymbol{V}^{-1} \boldsymbol{A} \boldsymbol{V} = \mathrm{diag}(\lambda_1, \ldots, \lambda_n)$ with $\boldsymbol{V} = \begin{bmatrix} \boldsymbol{v}_1 & \cdots & \boldsymbol{v}_n \end{bmatrix}$. Assume that $|\lambda_1| > |\lambda_2| \geq \cdots \geq |\lambda_n|$. The pseudocode for the power method is shown below (Golub & Van Loan, 2013):

---

**Algorithm 1:** Power method for finding the largest principal eigenvalue of the matrix $A$

---

**1** **Given** $\boldsymbol{A} \in \mathbb{R}^{n \times n}$ an $n \times n$ matrix, an arbitrary unit vector $x^{(0)} \in \mathbb{R}^n$, the maximum number of iterations $k_{\max}$, and the stopping criterion $\epsilon$.

**2** **for** $k = 1, 2, \ldots, k_{max}$ **do**

**3**      Compute $\boldsymbol{y}^{(k)} = \boldsymbol{A}\boldsymbol{x}^{(k-1)}$.

**4**      Normalize $\boldsymbol{x}^{(k)} = \frac{\boldsymbol{y}^{(k)}}{\|\boldsymbol{y}^{(k)}\|}$.

**5**      Compute the difference $\delta = \|\boldsymbol{x}^{(k)} - \boldsymbol{x}^{(k-1)}\|$.

**6**      **if** $\delta < \epsilon$ **then**

**7**          Record the largest principal eigenvalue using the Rayleigh quotient,

$$\lambda^{(k)} = \frac{\langle \boldsymbol{x}^{(k)}, \boldsymbol{A}\boldsymbol{x}^{(k)} \rangle}{\langle \boldsymbol{x}^{(k)}, \boldsymbol{x}^{(k)} \rangle}.$$

         The stopping criterion is met, the iteration can be stopped.

---

Let us examine the convergence properties of the power iteration. If

$$\boldsymbol{x}^{(0)} = a_1 \boldsymbol{v}_1 + a_2 \boldsymbol{v}_2 + \cdots + a_n \boldsymbol{v}_n$$

and $\boldsymbol{v}_1 \neq 0$, then

$$\boldsymbol{A}^k \boldsymbol{x}^{(0)} = a_1 \lambda_1^k \left( \boldsymbol{v}_1 + \sum_{j=2}^{n} \frac{a_j}{a_1} \left( \frac{\lambda_j}{\lambda_1} \right)^k \boldsymbol{v}_j \right).$$

Since $\boldsymbol{x}^{(k)} \in \mathrm{span}\{\boldsymbol{A}^k \boldsymbol{x}^{(0)}\}$, we conclude that

$$\mathrm{dist}\left( \mathrm{span}\{\boldsymbol{x}^{(k)}\}, \mathrm{span}\{\boldsymbol{v}_1\} \right) = O\left( \left( \frac{\lambda_2}{\lambda_1} \right)^k \right).$$

It is also easy to verify that

$$|\lambda_1 - \lambda^{(k)}| = O\left( \left( \frac{\lambda_2}{\lambda_1} \right)^k \right).$$

Since $\lambda_1$ is larger than all the other eigenvalues in modulus, it is referred to as the largest principal eigenvalue. Thus, the power method converges if $\lambda_1$ is the largest principal and if $\boldsymbol{x}^{(0)}$ has a component in the direction of the corresponding dominant eigenvector $\boldsymbol{x}_1$.

In practice, the effectiveness of the power method largely depends on the ratio $|\lambda_2|/|\lambda_1|$, as this ratio determines the convergence rate. Therefore, applying specific spectral transformations to the matrix to increase this ratio can significantly accelerate the convergence of the power method.

### A.2. Deflation Projection Details

Consider the scenario where we have determined the largest modulus eigenvalue, $\lambda_1$, and its corresponding eigenvector, $\boldsymbol{v}_1$, utilizing an algorithm such as the power method. These algorithms consistently identify the eigenvalue of the largest modulus from the given matrix along with an associated eigenvector. We ensure that the vector $\boldsymbol{v}_1$ is normalized such that

$\|\boldsymbol{v}_1\|_2 = 1$. The task then becomes computing the subsequent eigenvalue, $\lambda_2$, of the matrix $\boldsymbol{A}$. A traditional approach to address this is through what is commonly known as a deflation procedure. This technique involves a rank-one modification to the original matrix, aimed at shifting the eigenvalue $\lambda_1$ while preserving all other eigenvalues intact. The modification is designed in such a way that $\lambda_2$ emerges as the eigenvalue with the largest modulus in the adjusted matrix. Consequently, the power method can be reapplied to this updated matrix to extract the eigenvalue-eigenvector pair $\lambda_2, \boldsymbol{v}_2$.

When the invariant subspace requiring deflation is one-dimensional, consider the following Proposition A.1. The propositions and proofs below are derived from Saad (2011) P90.

**Proposition A.1.** *Let $\boldsymbol{v}_1$ be an eigenvector of $\boldsymbol{A}$ of norm 1, associated with the eigenvalue $\lambda_1$ and let $\boldsymbol{A}_1 \equiv \boldsymbol{A} - \sigma \boldsymbol{v}_1 \boldsymbol{v}_1^H$. Then the eigenvalues of $\boldsymbol{A}_1$ are $\tilde{\lambda}_1 = \lambda_1 - \sigma$ and $\tilde{\lambda}_j = \lambda_j, j = 2, 3, \ldots, n$. Moreover, the Schur vectors associated with $\tilde{\lambda}_j, j = 1, 2, 3, \ldots, n$ are identical with those of $\boldsymbol{A}$.*

*Proof.* Let $\boldsymbol{AV} = \boldsymbol{VR}$ be the Schur factorization of $\boldsymbol{A}$, where $\boldsymbol{R}$ is upper triangular and $\boldsymbol{V}$ is orthonormal. Then we have

$$\boldsymbol{A}_1 \boldsymbol{V} = \left[\boldsymbol{A} - \sigma \boldsymbol{v}_1 \boldsymbol{v}_1^\top\right] \boldsymbol{V} = \boldsymbol{VR} - \sigma \boldsymbol{v}_1 \boldsymbol{e}_1^\top = \boldsymbol{V}[\boldsymbol{R} - \sigma \boldsymbol{e}_1 \boldsymbol{e}_1^\top].$$

Here, $\boldsymbol{e}_1$ is the first standard basis vector. The result follows immediately. $\square$

According to Proposition A.1, once the eigenvalue $\lambda_1$ and eigenvector $\boldsymbol{v}_1$ are known, we can define the deflation projection matrix $\boldsymbol{P}_1 = \boldsymbol{I} - \lambda_1 \boldsymbol{v}_1 \boldsymbol{v}_1^\top$ to compute the remaining eigenvalues and eigenvectors.

When deflating with multiple vectors, let $\boldsymbol{q}_1, \boldsymbol{q}_2, \ldots, \boldsymbol{q}_j$ be a set of Schur vectors associated with the eigenvalues $\lambda_1, \lambda_2, \ldots, \lambda_j$. We denote by $\boldsymbol{Q}_j$ the matrix of column vectors $\boldsymbol{q}_1, \boldsymbol{q}_2, \ldots, \boldsymbol{q}_j$. Thus, $\boldsymbol{Q}_j \equiv [\boldsymbol{q}_1, \boldsymbol{q}_2, \ldots, \boldsymbol{q}_j]$ is an orthonormal matrix whose columns form a basis of the eigenspace associated with the eigenvalues $\lambda_1, \lambda_2, \ldots, \lambda_j$. An immediate generalization of Proposition A.1 is the following (Saad, 2011) P94.

**Proposition A.2.** *Let $\boldsymbol{\Sigma}_j$ be the $j \times j$ diagonal matrix $\boldsymbol{\Sigma}_j = diag(\sigma_1, \sigma_2, \ldots, \sigma_j)$, and $\boldsymbol{Q}_j$ an $n \times j$ orthogonal matrix consisting of the Schur vectors of $\boldsymbol{A}$ associated with $\lambda_1, \ldots, \lambda_j$. Then the eigenvalues of the matrix*

$$\boldsymbol{A}_j \equiv \boldsymbol{A} - \boldsymbol{Q}_j \boldsymbol{\Sigma}_j \boldsymbol{Q}_j^\top,$$

*are $\tilde{\lambda}_i = \lambda_i - \sigma_i$ for $i \leq j$ and $\tilde{\lambda}_i = \lambda_i$ for $i > j$. Moreover, its associated Schur vectors are identical with those of $\boldsymbol{A}$.*

*Proof.* Let $\boldsymbol{AU} = \boldsymbol{UR}$ be the Schur factorization of $\boldsymbol{A}$. We have

$$\boldsymbol{A}_j \boldsymbol{U} = \left[\boldsymbol{A} - \boldsymbol{Q}_j \boldsymbol{\Sigma}_j \boldsymbol{Q}_j^\top\right] \boldsymbol{U} = \boldsymbol{UR} - \boldsymbol{Q}_j \boldsymbol{\Sigma}_j \boldsymbol{E}_j^\top,$$

where $\boldsymbol{E}_j = [\boldsymbol{e}_1, \boldsymbol{e}_2, \ldots, \boldsymbol{e}_j]$. Hence

$$\boldsymbol{A}_j \boldsymbol{U} = \boldsymbol{U} \left[\boldsymbol{R} - \boldsymbol{E}_j \boldsymbol{\Sigma}_j \boldsymbol{E}_j^\top\right]$$

and the result follows. $\square$

According to Proposition A.2, if $\boldsymbol{A}$ is a normal matrix and the eigenvalues $\lambda_1, \ldots, \lambda_j$ along with their corresponding eigenvectors $\boldsymbol{v}_1, \ldots, \boldsymbol{v}_j$ are known, we can construct the deflation projection matrix $\boldsymbol{P}_j = \boldsymbol{I} - \boldsymbol{V}_j \boldsymbol{\Sigma}_j \boldsymbol{V}_j^\top$ to compute the remaining eigenvalues and eigenvectors. Here, $\boldsymbol{\Sigma}_j = \text{diag}(\sigma_1, \sigma_2, \ldots, \sigma_j)$ and $\boldsymbol{V}_j = [\boldsymbol{v}_1, \boldsymbol{v}_2, \ldots, \boldsymbol{v}_j]$.

### A.3. Filtering Technique

The primary objective of filtering techniques is to manipulate the eigenvalue distribution of a matrix through spectral transformations (Saad, 2011). This enhances specific eigenvalues of interest, facilitating their recognition and computation by iterative solvers. Filter transformation functions, $F(x)$, typically fall into two categories:

1. Polynomial Filters, expressed as $P(x)$, such as the Chebyshev filter (Miao & Wu, 2021; Banerjee et al., 2016).

2. Rational Function Filters, often denoted as $P(x)/Q(x)$, such as the shift-invert method (Van Beeumen, 2015; Watkins, 2007). Below we describe this strategy in detail.

**Shift-Invert Strategy**   The shift-invert strategy applies the transformation $(A - \sigma I)^{-1}$ to the matrix $A$, where $\sigma$ is a scalar approximating a target eigenvalue, termed as shift. This operation transforms each eigenvalue $\lambda$ of $A$ into $\frac{1}{\lambda - \sigma}$, amplifying those eigenvalues close to $\sigma$ in the transformed matrix, making them larger and more distinguishable (Watkins, 2007).

For instance, consider the power method, where the convergence rate is primarily governed by the ratio of the matrix's largest modulus eigenvalue to its second largest. Suppose matrix $A$ has three principal eigenvalues: $\lambda_1 = 10$, $\lambda_2 = 3$, and $\lambda_3 = 2$. Our objective is to compute $\lambda_1$, the largest eigenvalue. In the original matrix $A$, the convergence rate of the power method hinges on the spectral gap ratio, defined as:

$$\text{Spectral Gap Ratio} = \frac{\lambda_1}{\lambda_2} \approx 3.33$$

Applying the shift-invert transformation with $\sigma = 9.5$ strategically selected close to $\lambda_1$, the new eigenvalues $\mu$ are recalculated as:

$$\mu_i = \frac{1}{\lambda_i - \sigma}$$

This results in transformed eigenvalues:

$$\mu_1 = 2, \quad \mu_2 \approx -0.133, \quad \mu_3 \approx -0.125$$

Under this transformation, $\mu_1 = 2$ emerges as the dominant eigenvalue in the new matrix, with the other eigenvalues significantly smaller. Consequently, the new spectral gap ratio escalates to:

$$\text{New Spectral Gap Ratio} = \frac{2}{0.133} \approx 15.04$$

This enhanced spectral gap notably accelerates the convergence of the power method in the new matrix configuration.

Filtering techniques are often synergized with techniques like the implicit restarts of Krylov algorithms (Watkins, 2007; Golub & Van Loan, 2013), employing matrix operation optimizations to minimize the computational demands of evaluating matrix functions. These methods enable more precise localization and computation of multiple eigenvalues spread across the spectral range, particularly vital in physical (Salas et al., 2015; Banerjee et al., 2016) and materials science (Kohn, 1999) simulations where these eigenvalues frequently correlate with the system's fundamental properties (Winkelmann et al., 2019).

# B. Details of Experimental Setup

## B.1. Experimental Environment

To ensure consistency in our evaluations, all comparative experiments were conducted under uniform computing environments. Specifically, the environments used are detailed as follows:

- CPU: 72 vCPU AMD EPYC 9754 128-Core Processor

- GPU: NVIDIA GeForce RTX 4090D (24GB)

## B.2. Experimental Parameters

- NeuralSVD and NeuralEF: (Using the original paper settings)
  - Optimizer: RMSProp with a learning rate scheduler.
  - Learning rate: 1e-4, batch size: 128
  - Neural Network Architecture: layers = [128,128,128]
  - Laplacian regularization set to 0.01, with evaluation frequency every 10000 iterations.
  - Fourier feature mapping enabled with a size of 1024 and scale of 0.1.
  - Neural network structure: hidden layers of 128,128,128 using softplus as the activation function.
  - For the 1-dimensional problem, the number of points is $20,000$, with $400,000$ iterations. For the 2-dimensional problem, the number of points is $40,000 = 200 \times 200$, also with $400,000$ iterations. For the 5-dimensional problem, the number of points is $59,049 = 9^5$, with $500,000$ iterations.

- STNet

    - Optimizer: Adam
    - Learning rate: 1e-4
    - Neural Network Architecture: Assuming d is the dimension of the problem. For d = 1 or 2, layers = [d, 20, 20, 20, 20, 1] (For Harmonic operator d=2, layers = [d, 20, 20, 20, 1]). For d=5, layers = [d, 40, 40, 40, 40, 1]. For else case, layers = [d, 40, 40, 40, 40, 1].
    - For the 1-dimensional problem, the number of points is $20,000$, with $400,000$ iterations. For the 2-dimensional problem, the number of points is $40,000 = 200 \times 200$, also with $400,000$ iterations. For the 5-dimensional problem, the number of points is $59,049 = 9^5$, with $500,000$ iterations.

## B.3. Error Metrics

- Absolute Error:
  We employ absolute error to estimate the bias of the output eigenvalues of the model:

$$\text{Absolute Error} = |\tilde{\lambda} - \lambda|. \tag{17}$$

  Here $\tilde{\lambda}$ represents the eigenvalue predicted by the model, while $\lambda$ denotes the true eigenvalue.

- Residual Error:
  To further analyze the error in eigenpair $(\tilde{v}, \tilde{\lambda})$ predictions, we use the following metric:

$$\text{Residual Error} = ||\mathcal{L}\tilde{v} - \tilde{\lambda}\tilde{v}||_2. \tag{18}$$

  Here, $\tilde{v}$ represents the eigenfunction predicted by the model. When $\tilde{\lambda}$ is the true eigenvalue and $\tilde{v}$ is the true eigenfunction, the Residual Error equals 0.

## C. Analysis of Hyperparameters

**Model Depth**:

Table 7: Consider the 2-dimensional Harmonic problem, with the fixed layer width of 20, and compare the performance of STNet at different model layers. Other experimental details are the same as Appendix B.2.

| Layer | Index | $\lambda$ Absolute Error | Residual |
|---|---|---|---|
| 3 | $(v_1, \lambda_1)$ | 1.02e-5 | 4.56e-3 |
| | $(v_2, \lambda_2)$ | 3.04e-2 | 2.56e+1 |
| | $(v_3, \lambda_3)$ | 6.76e-2 | 6.99e+1 |
| | $(v_4, \lambda_4)$ | 1.00e-1 | 2.12e+3 |
| 4 | $(v_1, \lambda_1)$ | 1.42e-5 | 4.12e-3 |
| | $(v_2, \lambda_2)$ | 2.96e-1 | 1.24e+1 |
| | $(v_3, \lambda_3)$ | 4.17e-1 | 1.43e+1 |
| | $(v_4, \lambda_4)$ | 2.00e+1 | 2.17e+5 |
| 5 | $(v_1, \lambda_1)$ | 4.36e-6 | 4.12e-3 |
| | $(v_2, \lambda_2)$ | 8.63e-1 | 3.12e+1 |
| | $(v_3, \lambda_3)$ | 1.98e+0 | 1.58e+3 |
| | $(v_4, \lambda_4)$ | 8.94e+1 | 2.09e+3 |
| 6 | $(v_1, \lambda_1)$ | 1.06e-5 | 9.56e-3 |
| | $(v_2, \lambda_2)$ | 8.21e-1 | 2.00e+1 |
| | $(v_3, \lambda_3)$ | 1.17e+0 | 9.90e+3 |
| | $(v_4, \lambda_4)$ | 3.81e+1 | 7.53e+4 |

**Model Width**:

Table 8: Consider the 2-dimensional Harmonic problem, with the fixed layer depth of 3, and compare the performance of STNet at different model widths. Other experimental details are the same as Appendix B.2.

| Width | Index | $\lambda$ Absolute Error | Residual |
|---|---|---|---|
| 10 | $(v_1, \lambda_1)$ | 1.68e-6 | 1.26e-3 |
| | $(v_2, \lambda_2)$ | 3.82e-1 | 2.36e+0 |
| | $(v_3, \lambda_3)$ | 7.54e-1 | 1.20e+2 |
| | $(v_4, \lambda_4)$ | 1.71e-1 | 2.49e+3 |
| 20 | $(v_1, \lambda_1)$ | 1.42e-5 | 4.12e-3 |
| | $(v_2, \lambda_2)$ | 2.96e-1 | 1.24e+1 |
| | $(v_3, \lambda_3)$ | 4.17e-1 | 1.43e+1 |
| | $(v_4, \lambda_4)$ | 2.00e+1 | 2.17e+5 |
| 30 | $(v_1, \lambda_1)$ | 3.26e-5 | 2.25e-2 |
| | $(v_2, \lambda_2)$ | 1.50e+0 | 2.10e+1 |
| | $(v_3, \lambda_3)$ | 1.59e+0 | 8.21e+3 |
| | $(v_4, \lambda_4)$ | 3.52e+2 | 2.77e+5 |
| 40 | $(v_1, \lambda_1)$ | 1.57e-5 | 2.06e-2 |
| | $(v_2, \lambda_2)$ | 2.67e+0 | 5.03e+1 |
| | $(v_3, \lambda_3)$ | 7.93e+1 | 5.76e+3 |
| | $(v_4, \lambda_4)$ | 1.50e+2 | 1.47e+4 |

**The Number of Points**:

Table 9: Consider the 2-dimensional Harmonic problem and compare the performance of STNet at different numbers of points. Other experimental details are the same Appendix B.2.

| Number | Index | $\lambda$ Absolute Error | Residual |
|--------|-------|-----------------|----------|
| 20000 | $(v_1, \lambda_1)$ | 1.11e-5 | 3.19e-3 |
|  | $(v_2, \lambda_2)$ | 1.25e+0 | 3.22e+0 |
|  | $(v_3, \lambda_3)$ | 1.61e+0 | 1.27e+2 |
| 30000 | $(v_1, \lambda_1)$ | 4.40e-5 | 7.09e-3 |
|  | $(v_2, \lambda_2)$ | 3.58e-1 | 2.71e+0 |
|  | $(v_3, \lambda_3)$ | 1.70e-1 | 5.62e+1 |
| 40000 | $(v_1, \lambda_1)$ | 1.42e-5 | 4.12e-3 |
|  | $(v_2, \lambda_2)$ | 2.96e-1 | 1.24e+1 |
|  | $(v_3, \lambda_3)$ | 4.17e-1 | 1.43e+1 |
| 50000 | $(v_1, \lambda_1)$ | 4.94e-6 | 6.63e-3 |
|  | $(v_2, \lambda_2)$ | 2.53e-1 | 2.46e+1 |
|  | $(v_3, \lambda_3)$ | 3.73e-1 | 1.50e+3 |

The influence of model depth, model width, and the number of points on STNet is illustrated in Tables 7, 8, and 9, respectively. Experimental results indicate that STNet is relatively unaffected by changes in model depth and model width. However, it is significantly influenced by the number of points, with performance improving as more points are used.

