# OpenReview forum: "STNet: Spectral Transformation Network for Solving Operator Eigenvalue Problem"
_ICML.cc/2025/Conference — Submitted to ICML 2025_

### Official Review · Reviewer_h2hE · 2025-03-08

**Overall Recommendation:** 2

**Summary:**

The authors are interested in solving an eigenvalue problem for a differential operator.

**Claims And Evidence:**

The authors claim to approximate the eigenvalues with higher accuracy than competing approaches, which they do, but I am not sure about the overall performance as the competition seems really terrible.

**Essential References Not Discussed:**

There might be specific methods tailored for the PDE eigenvalue problems that the authors did not cite. All their references for EVPs are very generic, like books by Saad or Golub/van Loan.

**Experimental Designs Or Analyses:**

The authors compare for interesting eigenvalue problems and compare to other methods, albeit its not clear this is always done appropriately.

**Methods And Evaluation Criteria:**

The authors use interesting eigenvalue problems but since they are always giving absolute errors this could be misleading, also the performance shown in this measure fails to give two decently converged eigenvalues for a one-dimensional problem. Again the better results for higher dimensions could not be meaningful given that the errors are absolute.

**Other Comments Or Suggestions:**

Notation for the neural network as $NN$ confusing as in the next line $N$ appears as the number of sampling points.
Most of the references are formatted poorly. Capitalize the names of Krylov, Schur, Fokker, Planck, etc

**Other Strengths And Weaknesses:**

** Weakness**
For the computation of first few eigenvalues it would actually be possible to use sparse grids to obtain accurate results in 5 dimensions.  So if the goal is to compute first few eigenfunctions, the authors should compare this method against existing sparse grid methods to get high accuracy. Difficult to follow derivation of the suggested method and it is also unclear to me what the advantage is of a neural network based approach. I think the convergence behaviour is not well understood, which for the shift-and-invert method requires some spectral gap. This could have been exploited more. Comparison to outside of learning based approaches seems not sufficient (see sparse grid comment and in lower dimensions this should be able to give very accurate results).

**Questions For Authors:**

How does the method compare to other traditional schemes and the discretized eigenvalue problems.

**Relation To Broader Scientific Literature:**

The authors cite the generic numerical literature for eigenvalue problems.

**Theoretical Claims:**

There are no proofs in the manuscript.

---

### Official Review · Reviewer_6EVU · 2025-03-12

**Overall Recommendation:** 4

**Summary:**

This paper focus on solving eigenvalue and eigenfunction problem. Numerical methods suffer from the curse of dimensionality. There is a tread of attacking the problem with methods of deep learning, e.g. NeuralEF [1], NeuralSVD [2], etc. The authors try to improve the existing methods in terms precision. Their improvement is entirely based on two classic techniques in eigenvalue problems, i.e., deflation projection and filter transform. The implementation is straightforwardly summarized in Algorithm 2. These techniques are complementary to and thus can be combined with the work on neural network side.

Based on diversified experiments, the proposed method exhibited promising results that outperformed all baseline models on accuracy with the same number of iterations.

[1] Deng, Z., Shi, J., and Zhu, J. Neuralef: Deconstructing
kernels by deep neural networks. In International Conference
on Machine Learning, pp. 4976–4992. PMLR,
2022.

[2] Ryu, J. J., Xu, X., Erol, H., Bu, Y., Zheng, L., and Wornell,
G. W. Operator svd with neural networks via nested lowrank
approximation. arXiv preprint arXiv:2402.03655,
2024.

**Claims And Evidence:**

Yes.

**Essential References Not Discussed:**

N.A.

**Experimental Designs Or Analyses:**

Yes.

**Methods And Evaluation Criteria:**

Yes.

**Other Comments Or Suggestions:**

Typo:
1. Line 613, $\bf{A}_1=\bf{A}-\sigma \bf{v}_1 \bf{v}_1^H \rightarrow \bf{A}_1=\bf{A}-\sigma \bf{v}_1 \bf{v}_1^T$

**Other Strengths And Weaknesses:**

**Significance** The proposed method is of high value because of simplicity and effectiveness. It is purely based on classical techniques, e.g., deflation projection and filter transform, which is complementary to improvements on neural networks.


**Weakness**

1. In Section 2 Related Works,  the last paragraph seems a little irrelevant or the connection to the subject of this paper is not well elaborated.

2. The current experiment does not contain comparison wrt convergence speed. Would it be beneficial to present such comparison?

**Questions For Authors:**

What is the particular reason that in Table 1, NeuralEF does not have values for Dim=2 and Dim=5?

**Relation To Broader Scientific Literature:**

N.A.

**Theoretical Claims:**

I checked proofs in A.2 and A.3 and found no issues except a typo.

---

> ### Author Rebuttal · Authors · 2025-04-01
>
> We thank the reviewer for the insightful and valuable comments. We respond to your comments as follows and sincerely hope that our rebuttal could properly address your concerns. If so, we would deeply appreciate it if you could raise your score and your confidence. If not, please let us know your further concerns, and we will continue actively responding to your comments and improving our work.
>
> ## **Other Strengths And Weaknesses 1**
>
> > In Section 2 Related Works, the last paragraph seems a little irrelevant or the connection to the subject of this paper is not well elaborated.
>
> - Sorry for this problem. This problem will be corrected in future versions of the paper.
>
> ## **Other Strengths And Weaknesses 2**
>
> > The current experiment does not contain comparison wrt convergence speed. Would it be beneficial to present such comparison?
>
> - Thank you for your suggestions. Here’s experimental evidence to clarify our claims:
>   - 2D Harmonic Operator Case Study: We analyzed how NeuralSVD and STNet improve accuracy as iterations progress. The trend of absolute error reduction is shown in this figure: [Convergence Comparison](https://anonymous.4open.science/r/rebuttal2-534E/rebuttal1.3.pdf).
>
> - Key Takeaway: STNet achieves significantly faster convergence than NeuralSVD. This stems from STNet’s design, which mimics the power method’s iterative approach to eigenvalue estimation. We will add more detailed analysis and experiments in the final version of the paper.
>
> ## **Other Comments Or Suggestions**
>
> > Typo: Line 613
>
> - We apologize for the confusion caused. This error will be corrected in future versions of the paper.
>
> ## **Questions**
>
> > What is the particular reason that in Table 1, NeuralEF does not have values for Dim=2 and Dim=5?
>
> - As described on line 311 of the paper, NeuralEF encountered numerical instability in the 2D and 5D harmonic operator problems. The resulting errors were significantly larger than the target eigenvalues, making the data unsuitable for meaningful comparison. Thus, we excluded these results.
> - NeuralEF’s inability to accurately resolve eigenvalues with relatively small magnitudes may stem from stochastic optimization or numerical errors. This limitation is also noted in NeuralEF’s original paper ([2], page 9, Section 6, paragraph 2).
> - A similar issue is observed in NeuralSVD’s study. For the 2D harmonic oscillator experiment ([1], page 8, Figure 4b), NeuralEF exhibits a relative error of over 100%, which strongly aligns with our conclusions.
>
> [1] Operator SVD with Neural Networks via Nested Low-Rank Approximation, ICML 2024, https://github.com/jongharyu/neural-svd
>
> [2] NeuralEF: Deconstructing Kernels by Deep Neural Networks, ICML 2022

---

### Official Review · Reviewer_P3AG · 2025-03-13

**Overall Recommendation:** 3

**Summary:**

The paper introduces the Spectral Transformation Network for solving operator eigenvalue problems, addressing challenges posed by high-dimensional operators. STNet uses deflation projection to remove the subspace corresponding to already-computed eigenfunctions, ensuring that the network does not converge to the same eigenpair repeatedly and reducing the effective search space. STNet also uses filter transform to amplify eigenvalues in the target region while suppressing others, thereby increasing the spectral gap and accelerating convergence.

Experiments on the harmonic eigenvalue problem, the Schrodinger oscillator equation, and the Fokker–Planck equation demonstrate that STNet achieves SOTA accuracy, outperforming existing deep learning-based methods and traditional numerical approaches in high-dimensional settings.

## update after rebuttal

I thank the authors for the updated information. I maintain a positive score for this submission.

**Claims And Evidence:**

The claim that STNet achieves state-of-the-art accuracy in computing operator eigenvalues compared to existing deep learning and traditional methods is well supported by diverse experiments. STNet's performance improvement is significant.

The effectiveness of two components of STNet, deflation projection and filter transform, are both validated through ablation.

**Essential References Not Discussed:**

I don't notice any essential references are missing.

**Experimental Designs Or Analyses:**

The experimental designs and analyses in the paper are sound to me. They provide convincing evidence for the superior performance of STNet, with appropriate benchmarks and metrics.

**Methods And Evaluation Criteria:**

The baselines and evaluation criteria are appropriate for the operator eigenvalue problems at hand.

**Other Comments Or Suggestions:**

I recommend that the authors add visualizations to clearly illustrate the effects of both deflation projection and filter transform. This would greatly enhance the reader's intuition and understanding of these mechanisms.

**Other Strengths And Weaknesses:**

The paper presents a novel idea and is clearly written. The experiments show significant improvements.

**Questions For Authors:**

I have no additional questions.

**Relation To Broader Scientific Literature:**

There are no significant connections to broader scientific literature in my view.

**Theoretical Claims:**

There is no proof provided in the main paper. I did not verify the proofs in the appendix.

---

> ### Author Rebuttal · Authors · 2025-04-01
>
> We thank the reviewer for the insightful and valuable comments. We respond to your comments as follows and sincerely hope that our rebuttal could properly address your concerns. If so, we would deeply appreciate it if you could raise your score and your confidence. If not, please let us know your further concerns, and we will continue actively responding to your comments and improving our work.
> ## **Other Comments Or Suggestions**
>
> > I recommend that the authors add visualizations to clearly illustrate the effects of both deflation projection and filter transform. This would greatly enhance the reader's intuition and understanding of these mechanisms.
>
> - Thank you for this excellent suggestion. We have prepared preliminary visualizations that effectively demonstrate the spectral transformations achieved by these two key components:
>
> 1. **Deflation Projection** (shown in https://anonymous.4open.science/r/rebuttal2-534E/rebuttal1.1.pdf):
>    1. This operation maps already-computed eigenvalues to zero while preserving other target eigenvalues
>    2. The visualization clearly shows how this prevents solved eigenvalues from interfering with subsequent computations
> 2. **Filter** **Transform** (shown in https://anonymous.4open.science/r/rebuttal2-534E/rebuttal1.2.pdf):
>    1. This transformation modifies the spectral distribution by:1. Amplifying the region containing eigenvalues of interest. 2. Compressing less relevant spectral regions
>    2. The effect significantly improves the solvability of target eigenvalues
>
> - For the final version, we will refine these visualizations for greater clarity.

---

### Official Review · Reviewer_bdUG · 2025-03-19

**Overall Recommendation:** 2

**Summary:**

This paper proposes a method to find eigenfunctions of a given operator using neural networks.
The idea is to combine ideas from numerical linear algebra to train neural networks to fit underlying eigenufnctions: (1) power method, (2) deflation projection, and (3) filter transform. The experiments are performed for different differential operators.

## update after rebuttal

I appreciate the authors' response and I have increased my score to 2. I believe the empirical results are promising and interesting (including the significantly smaller number of parameters), assuming the experiments were performed as described in the response. That said, I find that several descriptions, such as those of the objective functions, implementations, and differences from existing methods, remain ambiguous and could be significantly streamlined to demonstrate its merit. In the case of acceptance or future submission, please carefully revise the manuscript to address these issues.

**Claims And Evidence:**

There are experiments for three PDEs, but the performance of baseline methods (especially NeuralEF and NeuralSVD) do not seem to be in the reasonable range; see below (Other Comments and Questions) for detailed comments.

**Essential References Not Discussed:**

References seem adequate.

**Experimental Designs Or Analyses:**

There are some flaws and issues in the experimental setup. See comment below.

**Methods And Evaluation Criteria:**

- Compared to the existing methods NeuralEF and NeuralSVD, the difference seems to be in the use of power method and the idea of filter transform, which is a nice addition to the learning framework.
- One thing unclear is the core difference of this STNet to the power method in (Yang et al., 2023).
- A similar idea of the deflation technique was proposed to be applied to NeuralEF and NeuralSVD in the terminology of "sequential nesting" (Ryu et al., 2024).
- Compared to NeuralEF or NeuralSVD, where the objective function is well defined and characterizes the desired ordered eigenfunctions in the order of eigenvalues, the objective function of STNet is not properly described. Eq. (8) is mentioned as the optimization problem, but it is not computable as it does not have access to the underlying eigenfunction $v_i$'s. It is then mentioned in the first paragraph of Section 4.2 that `Since neural networks cannot directly implement the inverse operator, we enforce ... through a suitable loss function.`, but there is no loss function defined other than the loss in line 10 of Algorithm 2. It is not properly explained how this loss function is derived.

**Other Comments Or Suggestions:**

- This sentence is incorrect: `This variability primarily arises because NeuralEF and NeuralSVD employ a uniform grid to acquire data points, whereas STNet uses uniform random sampling.` The NeuralEF paper did not consider the operator eigenvalue problem, and there is no reason to say that "NeuralEF employs a uniform grid". Also, the NeuralSVD paper proposed to draw fresh sample for every minibatch from a sampling distribution, which doesn't necessarily be uniform; see Appendix D.3 for the implementation with importance sampling and Appendix E.1.2 for that Gaussian distribution was used for harmonic oscillator in (Ryu et al., 2024). At a higher level, NeuralEF and NeuralSVD are optimization frameworks to find eigenfunctions, and they do not need to be necessarily associated with a specific sampling scheme. This makes the comparisons in the paper questionable.
- It is also unclear if STNet used "uniform random sampling" or "uniform grid", as the description in p.14 for STNet also indicates that something like `For the 1-dimensional problem, the number of points is 20, 000, ...`.

**Other Strengths And Weaknesses:**

On top of the incompleteness of the methodology part (the missing comparison of STNet to (Yang et al. 2023) and missing justification of the loss function), I have a serious concern about the experimental results, which are detailed below.

**Questions For Authors:**

- Why are the neural network architectures used for NeuralEF/NeuralSVD and STNet are different? Since all methods parameterize eigenfunctions, I think the most natural thing is to use same architecture throughout for a fair comparison.
- For a more informative comparison, when reporting accuracies of eigenvalue estimates, relative errors might be a better metric rather than absolute errors. Or, at least, indicate the true eigenvalues.
- How were the results for NeuralEF and NeuralSVD obtained? For example in Table 1, the results of NeuralEF and NeuralSVD are unreasonably bad even for dim=1,2, which is inconsistent to the fairly reasonable performance reported in the NeuralSVD paper for other operators. Results in Table 3 for harmonic oscillators is also inconsistent to what's reported in (Ryu et al., 2024); see Fig. 7(b).
- Why are the rows missing for NeuralEF in Table 1?

Given all these concerns, I believe that the experiments in this paper should be reexamined.
The pros and cons of the proposed framework should be also carefully explained in the methodology section, especially compared to NeuralEF and NeuralSVD, which are based on well-defined optimization problems.

**Relation To Broader Scientific Literature:**

Many scientific and ML problems can be formulated via operator eigenvalue problems, and thus solving them in an efficient manner is of great importance. Using neural networks can help circumvent the curse of dimensionality. Improved techniques for operator eigenvalue problems are desirable and can lead to important breakthroughs, especially in physical simulations.

**Theoretical Claims:**

There is no theoretical claim.

---

> ### Author Rebuttal · Authors · 2025-04-01
>
> We thank the reviewer for the insightful and valuable comments. We respond to your comments as follows and sincerely hope that our rebuttal could properly address your concerns. If so, we would deeply appreciate it if you could raise your score and your confidence. If not, please let us know your further concerns, and we will continue actively responding to your comments and improving our work.
> ## Methods  1
> The core difference between STNet and PMNN is that STNet uses Deflation Projection and Filter Transforming for better approximation:
>
> - Deflation Projection prevents STNet from converging to the predicted eigenfunctions, so that STNet can predict multiple eigenfunctions, while PMNN fails to do so. Also, the ablation study in Sec 5.5 of Deflation Projection demonstrates our contribution.
> - Filter Transforming is spectral transforming, improving STNet's convergence performance. And that's the reason why STNet outperforms PMNN. Also, we have the ablation study in Sec 5.5 to verify this contribution.
>
> STNet and PMNN are all based on power method, but STNet also employs the Deflation Projection and Filter Transform.
> ## Methods 2
> We argue that Deflation Projection is different from Sequential Nesting:
>
> - Different Ideas: The idea of Deflation Projection comes from power method, where we need a more powerful loss for iterative updating. While Sequatial Nesting takes eigenvalues problem as optimal problem, so they employ LoRA in Sequential Nesting serving for optimal problem.
> - Different Loss Functions: As stated in Eq 9 at Line 204, STNet approximates the power method, so the loss function here is norm of difference between functions. But NeuralSVD employs inner product of two functions for loss function.
> - Different Mathematical Purposes: **Deflation Projection performs spectral transforming** of operator problem, **which theoretically excludes the solved eigenfunction**. Sequential Nesting introduces LoRA for avoiding converging to the same eigenfunction.
>
> Generally, Deflation Projection is different from Sequential Nesting. To avoid this misunderstanding, we will add theoretical analysis to stress the difference.
> ## Methods3
> 1. Methodological Differences:
>    • NeuralEF/NeuralSVD reformulate eigenvalue problems as optimization tasks with custom loss functions.
>    • Our approach directly mimics the power method through neural updates, enhancing performance via spectral transformations rather than problem reformulation.
> 2. Clarification of Eq. 8:
>    • This equation represents STNet's theoretical objective (approximating target eigenfunctions), not a computational procedure.
> 3. Loss Function Implementation:
>    • The loss function design is detailed in Section 4.2 (Line 181).
>    • We will include a complete derivation in the final version for clarity.
> ## Comments
> 1. Yes, both NeuralSVD and NeuralEF use resampling for point selection. This was an oversight in our writing, and we will correct this section in the final version.
> 2. For our experiments, we used the official implementation of [1], following its default sampling settings in their paper. Importantly, this writing error does not affect the validity of our results. We sincerely apologize for the confusion.
> 3. For consistency, STNet uses random uniform sampling throughout all experiments. Specifically, we initialize with 20,000 randomly sampled points (as mentioned on lines 294 and 374), which are reused during iterations.
> ## Q1
> 1. The model architectures (depth and width) for [1] and [3] in our experiments were taken directly from the official code [1]. Similarly, the PMNN model parameters were adopted from its official implementation [2]. For STNet, we kept its architecture identical to PMNN to enable a fair comparison.
> 2. Parameter Efficiency: [1] requires ~100,000 parameters per eigenvalue-solving module, while STNet uses only 1,500 parameters per module. This further highlights STNet’s efficiency and strong performance despite its simplicity.
> ## Q2
> 1. In cases where eigenvalues can be zero (e.g., the Fokker-Planck Equation example on page 7, line 370), using the relative error formula becomes impossible (since dividing by zero is invalid). To maintain consistency, all eigenvalue errors in our experiments are reported using the absolute error.
>
> 1. For every operator, we explicitly provide the true eigenvalues before analyzing experimental results.
>    1. The Harmonic Eigenvalue Problem is listed on page 6, line 300
>    2. The Schrödinger Oscillator Equation is on page 9, line 326
>    3. The Fokker-Planck Equation is on page 10, line 346
> ## Q3
> - In all experiments, [3] and [1] were implemented using the official code from [1]'s GitHub repository. We only modified the target differential operator being solved. For the problems in Table 3, we strictly used the original code without any modifications. All reported results are authentic and reproducible.
> ## Q4
> - Please see the response to reviewer 6EVU **Questions**
>
> [1] NeuralSVD
>
> [2] PMNN
>
> [3] NeuralEF

---

### Decision · Program_Chairs · 2025-05-01

**Decision:**

Reject

**Comment:**

This paper studies the eigenvalue problems of some operators arising from PDEs. The paper proposes a new scheme, the Spectral Transformation Network (STNet). Comprehensive numerical experiments have been completed, though there is little theoretical investigation. Main concerns focus on lacking comparison against existing sparse grid methods, and convergence behaviour not well understood (Review h2hE). Given the lack of thorough theoretical grounding and insufficient benchmarking against strong classical baselines, the work does not meet the bar for acceptance.